# Pairing Binge Drinking and a High-Fat Diet in Adolescence Modulates the Inflammatory Effects of Subsequent Alcohol Consumption in Mice

**DOI:** 10.3390/ijms22105279

**Published:** 2021-05-17

**Authors:** Macarena González-Portilla, Sandra Montagud-Romero, Francisco Navarrete, Ani Gasparyan, Jorge Manzanares, José Miñarro, Marta Rodríguez-Arias

**Affiliations:** 1Department of Psychobiology, Facultad de Psicología, Universitat de Valencia, Avda. Blasco Ibáñez 21, 46010 Valencia, Spain; macarena.gonzalez@uv.es (M.G.-P.); jose.minarro@uv.es (J.M.); 2Department of Psychology and Sociology, University of Zaragoza, C/ Ciudad Escolar s/n, 44003 Teruel, Spain; sandra.montagud@unizar.es; 3Instituto de Neurociencias, Universidad Miguel Hernández-CSIC, Avda. de Ramón y Cajal s/n, San Juan de Alicante, 03550 Alicante, Spain; fnavarrete@umh.es (F.N.); agasparyan@umh.es (A.G.); jmanzanares@umh.es (J.M.); 4Red Temática de Investigación Cooperativa en Salud (RETICS-Trastornos Adictivos), Instituto de Salud Carlos III, MICINN and FEDER, 28029 Madrid, Spain

**Keywords:** binge drinking, alcohol, high-fat diet, binge, microbiota, cytokines, inflammation

## Abstract

Alcohol binge drinking (BD) and poor nutritional habits are two frequent behaviors among many adolescents that alter gut microbiota in a pro-inflammatory direction. Dysbiotic changes in the gut microbiome are observed after alcohol and high-fat diet (HFD) consumption, even before obesity onset. In this study, we investigate the neuroinflammatory response of adolescent BD when combined with a continuous or intermittent HFD and its effects on adult ethanol consumption by using a self-administration (SA) paradigm in mice. The inflammatory biomarkers IL-6 and CX3CL1 were measured in the striatum 24 h after BD, 3 weeks later and after the ethanol (EtOH) SA. Adolescent BD increased alcohol consumption in the oral SA and caused a greater motivation to seek the substance. Likewise, mice with intermittent access to HFD exhibited higher EtOH consumption, while the opposite effect was found in mice with continuous HFD access. Biochemical analyses showed that after BD and three weeks later, striatal levels of IL-6 and CX3CL1 were increased. In addition, in saline-treated mice, CX3CL1 was increased after continuous access to HFD. After oral SA procedure, striatal IL-6 was increased only in animals exposed to BD and HFD. In addition, striatal CX3CL1 levels were increased in all BD- and HFD-exposed groups. Overall, our findings show that adolescent BD and intermittent HFD increase adult alcohol intake and point to neuroinflammation as an important mechanism modulating this interaction.

## 1. Introduction

Alcohol is the most commonly abused substance worldwide and its consumption constitutes a public health concern and a tremendous economic burden [1]. Alcohol use is commonly initiated during adolescence by engaging in intermittent episodes of binge drinking (BD) alternated with abstinence periods [2]. This pattern of intake is defined as the consumption of at least five drinks for males and four drinks for females, reaching a blood alcohol concentration of 0.08 mg/dL or above in about a 2-h period [3]. BD has been associated with poor academic performance, psychosocial problems and physical injury [4,5]. In addition, an early onset of BD increases the risk for developing substance use disorders later in life, particularly alcohol use disorder (AUD) [6,7,8].

Adolescents and young adults are especially vulnerable to the deleterious consequences of BD due to the brain maturational processes that occur during this developmental period [9,10]. A large body of research has shown that BD during adolescence perturbs the structure and functioning of the prefrontal cortex and forebrain areas involved in reward [11,12,13]. These persisting alterations on brain regions are thought to underlie the long-term behavioral deficits associated with a greater vulnerability to develop an AUD [14,15,16].

In the last few years, the gut-brain axis has been highlighted as an important modulator of brain function and behavior [17]. Although the gut microbial community is sensitive to many factors, diet has been shown to be the primary factor shaping microbial diversity and composition [18]. Due to the ever-growing poor nutritional habits, the effects of an increased intake of food rich in both saturated fats and sugar on the microbiota is a subject of utmost concern. The consumption of a high-fat diet (HFD) is a dietary condition that changes the intestinal microbiome communities, even before the onset of obesity [19]. The gut microbiota can have indirect and direct effects in the central nervous system by acting on many pathways, primarily the regulation of inflammatory mediators [20]. Interestingly, clinical and experimental studies have reported elevated levels of inflammatory cytokines—(interleukin-6 (IL-6), interleukin-1beta (IL-1b), tumor necrosis factor alpha (TNFa)—associated with obesity or HFD consumption. Animal models of diet-induced obesity have demonstrated that consumption of HFD induces a central pro-inflammatory response as early as 3 days after [21,22]. Increased inflammatory signaling has been observed in areas in the brain such as the hypothalamus, amygdala, hippocampus and the cerebellum. Interestingly, these biomarkers of central inflammation correlated with cognitive deficits and anxiety symptomatology [23,24].

In parallel, alcohol consumption has been associated with dysbiotic changes in gut microbiome, breakdown of the intestinal barrier integrity and increased permeability of the intestinal mucosa [25,26]. One important source of brain damage and neurodegeneration caused by alcohol use is neuroinflammation [27,28,29,30,31]. Individuals with AUD exhibit elevated levels of several plasma cytokines such as IL-6, IL-10 and TNFa. In both clinical and preclinical studies, chronic—but also acute—exposure to alcohol is sufficient to elicit an increase in cytokine release [32,33,34]. Most importantly, blocking the alcohol-induced neuroimmune response by genetic elimination of TLR4 has prevented some long-term behavioral impairments [35]. Furthermore, the involvement of the immune response in modulating the neuropathological consequences induced by adolescent BD has also been described [36,37,38].

Epidemiological studies have revealed a high comorbidity between AUD and overeating pathologies such as obesity and binge-eating disorder (BED) [39]. Many obese and BED patients have difficulties restraining from consuming palatable foods (high in fat, sugar and/or salt content) and engage in compulsive behaviors similar to those observed in substance abuse disorder [40,41]. Compulsive overeating and binge eating are two frequent features of hedonic eating facilitated by poor impulse control. Binge eating is characterized by excessive intake of highly palatable food in a short time frame accompanied by a subjective sense of loss of control [42]. Overeating and binge eating are not only present in patients of eating disorders, but are also common in the general population, especially among adolescents [43]. In addition to behavioral commonalities, the neurobiological mechanisms modulating the reinforcing properties of drugs of abuse and palatable food are largely the same [44,45]. Both alcohol and HFD consumption activate the mesocorticolimbic reward circuitry consisting of dopaminergic neurons projecting from the midbrain (ventral tegmental area) to the nucleus accumbens in the striatum and the prefrontal cortex [46,47].

Animal models have been useful to study how palatable food influences dopamine signaling. It is now well characterized that intermittent access to HFD induces binge eating-like behavior in rodent models [48,49]. These studies have shown that food reward increases dopaminergic signaling in the striatum and that palatable food intake is specifically regulated by the nucleus accumbens [50,51]. Moreover, using optoinhibition techniques, it has been established that ventral striatal projections mediate binge eating behavior [52]. Evidence suggests that bingeing on HFD produces neuroadaptations similar to those of drugs of abuse [47,53]. Once exposed to HFD, mice brain reward circuits remain sensitized long after a return to a standard diet [54]. In this sense, consumption of HFD during adolescence may be a risk factor for the onset and escalation of excessive alcohol consumption in adulthood. The few studies that have explored how HFD bingeing impacts ethanol consumption have yielded diverging results depending on the access schedule used [55,56]. Some of these studies reported increases in alcohol intake after intermittent exposure to HFD [57,58], while other works observed a decrease in alcohol consumption following intermittent HFD [59,60]. The majority of studies have not included a continuous access HFD group to distinguish the possible unique effects of HFD bingeing on ethanol intake.

The purpose of this study is to further investigate the interaction between BD and two different access schedules of HFD during adolescence and its association to inflammatory markers and alcohol drinking in adulthood. Considering that both HFD and ethanol induces neuroinflammatory processes in overlapping brain circuits, we measured two inflammatory markers in the striatum when their consumption was combined. The implication of cytokine IL-6 and chemokine fractalkine (CX3CL1) gene expression in the central alcohol-related immune response made these two inflammatory mediators our choice [61,62,63].

In order to distinguish between the short-term and long-term effects, we evaluated the effects of BD in combination with HFD by measuring the IL-6 and CX3CL1 cytokine levels at three different time points. In this way, the study aimed to evaluate (1) the acute inflammatory effects of BD and intermittent or continuous HFD, (2) the long-term inflammatory effects of BD and intermittent or continuous HFD, and (3) the effects of BD and HFD during adolescence in increasing ethanol intake using the self-administration (SA) procedure and its association with inflammatory markers.

## 2. Results

### 2.1. Bodyweight Is Increased by Continuous but Not Intermittent Access to HFD

In Experiment 1, results obtained in the statistical analyses of body weight revealed an effect of the variable Weeks [F (9,387) = 624.07; *p* < 0.001], as mice gained weight every week as a consequence of growth. No main effect of diet on body weight was detected, showing that bingeing did not increase body weight. From week 6 onwards, all mice lost weight due to the self-administration (SA) deprivation (Figure 1A).

In Experiment 2, results obtained in the statistical analyses of body weight revealed an effect of the variable Weeks [F (9,396) = 567.389; *p* < 0.001], Diet [F (1,44) =7.701; *p* < 0.01] and the interaction Diet × Weeks [F (9,396) =1.127; *p* < 0.05]. Although all animals gained weight across weeks (*p* < 0.001), from week 3 onwards, mice in the HFDc-S and HFDc-E groups showed an increased body weight with respect to the groups fed with the standard diet (SD-S and SD-E) (Figure 1B).

### 2.2. Intermittent Access to HFD Induces Bingeing Behavior and Continuous Access Increases Caloric Intake

An escalation in the intake of the HFD was confirmed by the ANOVA. The caloric intake during binge sessions revealed an effect on the variable Weeks [F (9,180) = 199.377; *p* < 0.001]. The 2 h/kcal intake in session 5 was greater than session 1 and 2 (*p* < 0.05) (Figure 2A). At week 6, when the SA procedure began, kcal intake was greater due to the food restriction (*p* < 0.001).

Regarding the energy consumption of mice before the ethanol SA, the ANOVA revealed an effect of the variable Diet [F (2,18) = 56.165; *p* < 0.001]. The weekly energy intake was higher in the HFDc-S and HFDc-E groups compared to the rest of the groups (*p* < 0.001 for both groups fed on SD and both groups on HFDb) (Figure 2B).

### 2.3. A High Fat Diet and Intermittent Ethanol Intake during Adolescence Increases the Neuroinflammatory Response

After the end of the ethanol BD, the ANOVA of the striatal levels of IL-6 (Figure 3A) showed an effect of the variable Diet [F (2,57) = 7.680; *p* = 0.001], Treatment [F (1,57) = 31.752; *p* = 0.001], and the interaction Diet × Treatment [F (2,57) = 3.161; *p* = 0.05]. Intermittent and repeated ethanol administration during adolescence increased the striatal levels of IL-6 in mice fed with HFD (either in binge or continuously) with respect to those non-exposed to ethanol (*p* < 0.001 in both cases). Equally, HFDc-E and HFDb-E groups presented higher striatal levels of IL-6 than mice exposed to ethanol during adolescence but fed with the standard diet (SD-E) (*p* < 0.001 in both cases).

Three weeks after ethanol binge drinking, the ANOVA of the striatal levels of IL-6 (Figure 3B) showed an effect of the variable Treatment [F (1,51) = 9.332; *p* = 0.004]. All mice exposed to ethanol during adolescence showed higher levels of IL-6 (*p*< 0.01).

In groups exposed to food deprivation during ethanol SA, the ANOVA of the striatal levels of IL-6 (Figure 3C) showed an effect of the variable Diet [F (1,33) = 4.308; *p* = 0.046], Treatment [F (1,33) = 7.128; *p* = 0.012], and the interaction Diet × Treatment [F (1,33) = 4.278; *p* = 0.047]. Mice exposed to ethanol during adolescence and fed on HFD intermittently (HFDb-E) showed higher striatal levels of IL-6 after ethanol SA than the rest of the groups (*p* < 0.01 in all cases).

In groups that performed ethanol SA without food deprivation, the ANOVA of the striatal levels of IL-6 (Figure 3D) showed an effect of the variable Treatment [F (1,47) = 7.438; *p* = 0.009]. Groups exposed to ethanol during adolescence showed higher IL-6 levels (*p* < 0.01), although the effect was mainly due to the group fed on continuous HFD.

The ANOVA for the striatal levels of CX3CL1 after ethanol BD (Figure 4A) showed an effect of the variable Treatment [F (1,80) = 42.977; *p* = 0.001] and the interactions of Diet × Treatment [F (2,80) = 6.414; *p* = 0.003]. Mice fed on HFD continuously (HFDc-S) presented higher striatal level of CX3CL1 than those fed on the standard diet (*p* < 0.01) or intermittent HFD (*p* < 0.05). Ethanol exposure during adolescence increased CX3CL1 in all groups (*p* < 0.001 in all cases).

Similar results revealed the ANOVA for the striatal levels of CX3CL1 three weeks after the ethanol BD with an effect of the variable Treatment [F (1,54) = 39.365; *p* = 0.001], Diet [F (2,54) = 3.627; *p* = 0.033], and the interactions of Diet × Treatment [F (2,54) = 3.799; *p* = 0.029], (Figure 4B). Mice fed on HFD continuously (HFDc-S) presented higher striatal level of CX3CL1 than those fed on the standard diet or HFDb-S (*p* < 0.01 in both cases). Ethanol exposure during adolescence increased CX3CL1 in all groups (*p* < 0.001 in all cases).

In groups exposed to food deprivation during ethanol SA, the ANOVA of the striatal levels of CX3CL1 (Figure 4C) showed an effect of the variable Treatment [F (1,35) = 41.498; *p* = 0.001], and the interaction Diet × Treatment [F (1,35) = 9.504; *p* = 0.004]. Mice exposed to ethanol during adolescence showed higher striatal levels of CX3CL1 after ethanol SA than those treated with saline (*p* < 0.001 for SD-S and *p* < 0.05 for HFDb-S). However, those intermittently fed with HFD (HFDb-S) showed higher CX3CL1 levels than those fed only with the standard diet (*p* < 0.01).

In groups that performed ethanol SA without food deprivation the ANOVA of the striatal levels of CX3CL1 showed an effect of the variable Treatment [F (1,53) =23.123; *p* = 0.001], and the interaction Diet × Treatment [F (1,53) =9.421; *p* = 0.003] (Figure 4D). Ethanol exposure during adolescence induced higher CX3CL1 levels (*p* < 0.001 in all cases). The group fed on continuous HFD and treated with saline also showed increased levels of CX3CL1 with respect to SD-S (*p* < 0.001).

### 2.4. Oral Self-Administration of Ethanol

#### 2.4.1. Experiment 1

The ANOVA for the EtOH consumption (g/kg) during the FR1 schedule revealed a significant effect of the variable Treatment [F (1,46) = 9.860; *p* < 0.01] and the interaction Days × Diet [F (4,184) = 2.503; *p* < 0.05] (Figure 5A). Animals that received BD treatment during adolescence (SD-E and HFDb-E) exhibited increased alcohol consumption compared to those that received saline injections (SD-S and HFD-S) (*p* < 0.01). Also, mice in the HFDb groups (both HFDb-E and HFDb-S) exhibited an increased ethanol oral SA of EtOH (6%) in day 1 compared to day 2 (*p* < 0.01), day 3 (*p* < 0.05) and day 4 (*p* < 0.01). When both saline groups (SD-S and HFDb-S) were independently analyzed in order to confirm the effect of HFD in ethanol intake, the ANOVA revealed a significant effect of the variable Diet [F (1,24) = 6.065; *p* < 0.05]. Mice in the HFDb-S group exhibited increased ethanol consumption compared to the SD-S group (*p* < 0.05). With respect to the number of effective responses during FR1, the ANOVA revealed a significant effect of the variable Treatment [F (1,47) = 5.950; *p* < 0.05], (Figure 5B). Mice who received ethanol treatment (SD-E and HFDb-E) performed more effective responses compared to those who received saline (*p* < 0.05).

During the FR3 schedule, the ANOVA for the ethanol consumption revealed a significant effect of the interaction Days × Diet × Treatment [F (4,184) = 6.953 *p* < 0.001] (Figure 5A). Post-hoc comparisons showed that animals in the HFDb-E group presented increased ethanol intake compared to the HFDb-S group on days 7 (*p* < 0.05), 8 (*p* < 0.001), 9 and 10 (*p* < 0.5 for both). Moreover, animals in the SD-E group exhibited an increased alcohol consumption compared to the SD-S group on days 9 and 10 (*p* < 0.05 for both). With respect to effective responses, the ANOVA revealed a significant effect of the interaction Days × Diet × Treatment [F (4,184) =4.223; *p* < 0.01] (Figure 5B). Mice in the SD-E group performed more effective responses than the SD-S on days 9 and 10 (*p* < 0.05 for both). Similarly, animals in the HFDb-E group performed more effective responses than the HFD-S group on days 7, 8 (*p* < 0.5) and 10 (*p* < 0.001).

Analyses of the PR showed an effect of the variable Treatment [F (1,42) = 16.456; *p* < 0.001] in the motivation to seek alcohol (Figure 5C). Animals that received BD treatment during adolescence (SD-E and HFDb-E) presented higher breaking point values compared to the saline groups (*p* < 0.001). In reference to ethanol consumption, results showed an effect of the interaction Diet × Treatment [F (1,42) = 4.194; *p* < 0.05], (Figure 5D). Animals on the SD-E group exhibited an increased ethanol consumption than animals in the SD-S group (*p* < 0.05).

#### 2.4.2. Experiment 2

For EtOH consumption (g/kg), the ANOVA of the FR1 schedule revealed a significant effect of the interaction Days × Diet × Treatment [F (4,164) = 2.658 *p* < 0.05], (Figure 6A). On day 1, mice in the SD-E group exhibited an increased ethanol oral SA of EtOH (6%) with respect to the SD-S group (*p* < 0.05). With respect to the number of effective responses during FR1, the ANOVA revealed a significant effect of the variable Treatment [F (1,41) = 7.136; *p* < 0.05] (Figure 6B). Animals that received BD treatment during adolescence (SD-E and HFDc-E) presented increased effective responses with respect to the saline groups (*p* < 0.05).

The ANOVA for the ethanol consumption during FR3 schedule revealed a significant effect of the interaction Days × Diet × Treatment [F (4,156) = 3.853 *p* < 0.01] (Figure 6A). Mice in the SD-E group showed an increased ethanol consumption rate compared to those in the SD-S group on days 9 and 10 (*p* < 0.05 for both). With respect to effective responses, the ANOVA revealed a significant effect of the interaction Days × Treatment [F (4,172) = 3.126; *p* < 0.05] (Figure 6B). Animals that received BD treatment presented more effective responses with respect to the saline groups on day 9 (*p* < 0.01).

Analyses of the PR revealed no differences in the motivation to seek alcohol between groups (Figure 6C). In reference to ethanol consumption, results showed an effect of the variable Diet [F (1,44)= 9.486; *p* < 0.01], (Figure 6D). Animals on the SD group (SD-E and SD-S) exhibited an increased ethanol consumption than animals in the HFDc group (HFDb-E and HFDc-S) (*p* < 0.01).

## 3. Discussion

BD during adolescence has been associated with an increased alcohol drinking in adulthood concomitant with a neuroimmune response characterized by the activation of glia immunity receptors, an increased TLR4 signaling and a greater presence of inflammatory mediators and cytokines or chemokines [7,8,13]. Due to the high frequency of BD and high-fat food consumption among adolescents [64,65] we aimed to test the effects of HFD consumption on two important associated consequences of early BD: the neuroinflammatory response and the increased alcohol drinking in adult mice.

We obtained several key findings from this study. Firstly, we confirmed that ethanol exposure in a BD pattern during adolescence increased ethanol intake in adulthood. Secondly, we observed that different access schedules to HFD modulate the self-administered alcohol consumption in adulthood. Mice that binged on HFD three times a week increased ethanol consumption during the SA, regardless of also being exposed to BD (HFDb-E), although only this group exhibited a higher motivation to seek alcohol during the PR session. On the other hand, mice with continuous access to HFD decreased their alcohol consumption and exhibited lower breaking points during the PR session. Thirdly, despite the differential effect of a continuous or intermittent access to HFD on ethanol intake, we confirmed that BD and/or HFD consumption during adolescence induces a neuroinflammatory response manifested on the increased levels of IL-6 and CX3CL1. Striatal IL-6 and CX3CL1 levels were increased immediately after the BD procedure during adolescence and persisted elevated three weeks later. Moreover, continuous access to HFD (HFDc) also increased CX3CL1 in saline-treated mice. After the SA procedure, striatal IL-6 levels were increased only in mice exposed to both BD and HFD, while CX3CL1 proved to be more sensitive and was increased in the groups exposed to either BD, HFD or both together.

### 3.1. Access Schedule to HFD Exposure Differentially Affects the Increase in Ethanol Intake Induced by BD Exposure during Adolescence

In the present study we used a validated protocol of intraperitoneal EtOH injections to mimic the effects of alcohol binge drinking in adolescents [66,67]. The experimental design of our study focused on discerning the differential effects of intermittent or continuous access to HFD on alcohol drinking. Accumulative evidence has shown that HFD is a dietary condition with deep impact on the gut microbial composition and thereby, an important modulator of brain function [68,69]. Presumably, different HFD dietary compositions and intake patterns exert different effects on peripheral systems and central signaling [70]. For instance, we have described two different behavioral profiles in mice that had intermittent or continuous access to HFD during adolescence [56]. In addition, a recent study has described a unique gut microbial profile in BED patients compared to obese individuals [71].

To model binge eating in the HFDb groups mice had intermittent 2-h access to HFD three times per week. Mice in the HFDb group gradually increased the fat intake during the binge sessions, thus confirming the emergence of binge-like behavior. In line with previous reports, bodyweight in HFDb mice did not increase with respect to the SD groups. As expected, prolonged exposure to HFD in the HFDc groups induced an increase in average kilocalorie intake and thus, mice exhibited increased body weight compared to the HFDb and SD groups.

After reaching adulthood (PND 61), using the SA paradigm, we studied alcohol-related behaviors in mice, which reflect various AUD components in humans [72]. Although food and/or water deprivation is a common approach to facilitate operant learning in the SA procedure, non-deprived SA protocols are increasingly common [73,74,75,76]. In Experiment 1, when the SA procedure started, mice only had access to food for 1 h. Because our study aimed to differentiate between the effects of continuous HFD and intermittent bingeing on HFD, limiting the food access during the SA presented apparent disadvantages. Restricting the food access to 1 h in the HFDc group would have resembled the feeding schedule of the bingeing procedure. By conducting a deprived and a non-deprived SA in Experiments 1 and 2, respectively, we clearly distinguished the pattern of consumption of HFD in the HFDb and HFDc groups. In this sense, the lack of any food and water deprivation during the SA in Experiment 2 is seen as a significant advantage to compare the two HFD feeding conditions on alcohol consumption.

Our results confirmed that exposure to alcohol during the adolescence increases the alcohol SA consumption. In Experiment 1, mice that received BD treatment consumed more alcohol and exhibited a greater number of effective responses than those that received saline treatment under the FR1 schedule. These data are in agreement with previous findings indicating that exposure to alcohol during adolescence increases ethanol intake in later stages of life [76,77,78,79]. In addition, we also observed higher PR breaking point values in the animals that received BD treatment during adolescence. The PR ratio is a more demanding test that reflects the motivational aspect of drug-seeking, since the animal has to work harder to obtain the reinforcer [80]. Similarly, in Experiment 2 we observed an increase in the number of effective responses of mice that received BD treatment during adolescence (SD-E and HFDc-E), proving that the SA procedure without food deprivation is equally sensitive to detect these long-lasting effects. Therefore, our results confirm that prior exposure to alcohol is a strong predictor of alcohol drinking, particularly during adolescence.

We also noted that bingeing on fat produced an increase in alcohol drinking. The HFDb-S group exhibited an increased alcohol consumption compared to the SD-S group. Previous experiments in our laboratory showed that ethanol consumption is increased by HFD bingeing during adolescence [58,81]. The repeated stimulation of the reward system during episodes of HFD bingeing influences the dopaminergic signaling [82,83,84,85]. In this way, HFD bingeing during adolescence could have a cross-sensitization effect that would lead to differential behavioral responses to drugs’ rewarding properties in adulthood [86]. In a recent study, Mazzone et al. [54] proved that the HFD-related sensitization in the mesolimbic dopamine system persists long after restoration to a standard diet.

Results from Experiment 2 showed that the continuous access to HFD attenuated the increased self-administered alcohol consumption caused by early exposure to BD. Alcohol is a drug with caloric value (7 kcal/g) and has been shown to modulate food intake and body weight by acting on hypothalamic circuits modulating energy homeostasis [87,88]. Absorption, metabolism, caloric intake and satiety are some components of feeding behavior that are also important modulators of oral alcohol SA. Due to the caloric contribution of ethanol, metabolic pathways should also be considered in addition to the main reward pathways that drive alcohol consumption. In this sense, it is important to remark that the reduction in alcohol drinking may be attributable to a permanent satiated state in animals with an obese phenotype. In reference to substance reward, recent findings have shown that satiety signals can reduce the drive to seek SA of several drugs including cocaine and alcohol [89,90,91]. Furthermore, the hypothalamic and endocrine signals can decrease the activity of the dopaminergic system [92]. However, another possible explanation of these results is that long-term HFD consumption caused a general attenuation of dopamine signaling that may result in reduced alcohol drinking [93,94,95]. Exposure to HFD reduces dopamine receptor D2 protein expression levels in the striatum and leads to changes in dopamine synthesis and uptake [96,97,98,99]. In addition, chronic overconsumption of HFD can lead to reduced dopamine receptor D1 signaling, especially in the case of saturated lipids [100]. In the control group (SD-S), fed with the standard diet and not exposed to ethanol during adolescence, the consumption of alcohol and the number of effective responses was similar in the first and in the second experiments, pointing to a lack of satiety effect on ethanol intake under basal conditions.

### 3.2. HFD Increases the Neuroinflammatory Response Induced by Exposure to Binge Drinking during Adolescence

AUD is a condition associated with neuroadaptations, with morphological and immune alterations in neurons and glial cells that constitute the reward circuitry [101,102,103]. Changes in the inflammatory profile caused by alcohol have been described, and immune cells and cytokines are now regarded as biomarkers of alcohol use [10,104]. In the present study, we examined the IL-6 and CX3CL1 levels in the striatum, a key reward brain area, to distinguish between the acute and the long-term neuroinflammatory effects of BD and HFD during adolescence.

Several studies using different models of BD have shown that intermittent alcohol exposure activates an immune response in reward-related brain areas [66]. The inflammatory profile induced by adolescent BD has been shown to be a core contributor to the long-term alcohol-induced brain damage underlying the behavioral alterations and increased risk for AUD in adulthood [37,105,106]. Rodent studies have reported increased astrocytes and microglia production of IL-6 protein in the CNS after ethanol exposure [62,107,108], even after one single acute ethanol challenge [109]. In agreement with previous results [27,35,66], binge-like EtOH treatment during adolescence increased IL-6 levels in the striatum and this effect persists during adulthood. Biochemical analyses have shown increased striatal IL-6 levels 24 h after BD only when combined with a HFD, administered either intermittently or continuously (HFDb-E and HFDc-E). The combination of BD with HFD synergistically increased the IL-6 protein expression in the striatum. Moreover, three weeks later, IL-6 levels remained elevated in the same groups, but also in mice exposed to ethanol but fed with the standard diet (SD-E).

Besides being systematically increased in AUD patients, IL-6 is also elevated in obese subjects [61,110]. Long-term HFD feeding causes a microbiota modification, changing intestinal permeability and increasing bacterial lipopolysaccharide (LPS) [111]. Altogether, these substrates favor the onset of systemic inflammation upon the activation of TLR4 signaling pathway [112,113]. HFD-induced increased expression of proinflammatory cytokines can be observed as soon as one day [21] or even hours after a high-fat meal [114]. With our diet schedule, IL-6 did not show increases in mice fed with HFD without previous exposure to ethanol. Nevertheless, IL-6 changes highlight the additive effect of ethanol binge drinking and HFD. The observation that ethanol exerts a prolonged inflammatory response that persists into adulthood implies that BD during adolescence causes a long-lasting alteration of the neuroimmune function. The specific effects of increased IL-6 activity on behavior remain elusive but evidence from studies with genetically modified mice suggests that IL-6 is involved in the regulation of anxiety-like and depressive-like behaviors [108,115,116,117]. In this way, IL-6 inflammatory pathways may affect shared behavioral outcomes by ethanol and HFD consumption (anxiety, depression, craving) [118].

CX3CL1 is an adipocyte-derived inflammatory chemokine which is also up-regulated in obesity and during inflammatory processes in the CNS [119]. CX3CL1 is widely expressed in neurons located in forebrain structures involved in addiction, such as the amygdala, the prefrontal cortex and the striatum [120]. Among other physiological functions, CX3CL1 has been implicated in the regulation of cellular migration, synaptic pruning and the experience-dependent remodeling of circuits during normal and pathological brain development [121]. Human and animal studies have observed increased CX3CL1 expression in obesity [122,123] and in alcohol BD [63]. In our study, CX3CL1 was revealed to be an inflammatory marker highly sensitive to alcohol and HFD intake. We observed increased striatal levels of CX3CL1 immediately after BD in all feeding conditions (SD-E, HFDb-E and HFDc-E), as well as in saline-treated mice in the group continually fed with HFD (HFDc). Similar results in CX3CL1 levels were obtained three weeks after the last ethanol injection. Our results point to the fact that CX3CL1 may display a higher sensitivity to fat intake in obese animals. Available evidence supports a role of CX3CL1 during the early inflammatory processes observed in experimental obesity and metabolic disease [124,125]. Long-chain saturated fatty-acids are capable of activating the TLR4 signaling and trigger the expression of several chemokines such as CX3CL1 [126]. The lipidic contribution to the dietary macronutrient composition of mice in the HFDb group could not be significant to elicit an increase in striatal CX3CL1 levels.

Previous reports have also shown increased pro-inflammatory signaling and glia activation when intragastric chronic ethanol exposure is combined with HFD, although the experimental conditions are not comparable [127,128]. In both studies, HFD was administered for several weeks and ethanol was delivered intragastrically at doses that induce liver fibrosis. In this study, although other neuroinflammatory markers were elevated, shown by an increase in the gene expression of IL-1B, TNF-alpha and CX3CL1, the gene expression of IL-6 only increased after 8 weeks of exposure to both ethanol and HFD [128] but was not affected with a shorter exposure [127].

Chronic alcohol abuse in humans is associated with increases in serum inflammatory cytokines such as TNF and IL-6, which correlate with cognitive deterioration [129]. However, ELISA analyses after the SA procedure showed an increase in striatal IL-6 only in the animals that received BD treatment and HFD bingeing in adolescence (Experiment 1) and a subtler increase in the HFDc-E group (Experiment 2). Our results point to the fact that the neuroinflammatory response may be potentiated in animals previously exposed to alcohol that also consumed HFD. At this point, the acute neuroinflammatory response caused by BD is no longer observable, even after further ethanol intake during SA procedure. The inflammatory effects of ethanol depend on several factors including the length of the treatment, the maximum peak blood EtOH concentration, route of administration, age and sex [38]. In contrast to other paradigms, ethanol intake during the oral SA procedure is gradual and does not raise blood ethanol levels to such a large extent compared to binge paradigms or models of chronic EtOH exposure. For example, Schneider and co-workers [130] observed IL-6 increases after 30 days of 2 g/kg/day alcohol intake by oral gavage, which is a much higher concentration than that obtained with our SA procedure.

Striatal CX3CL1 levels after the SA procedure were increased in the groups that received BD treatment (SD-E, HFDb-E and HFDc-E). Moreover, the groups fed with HFD but not exposed to ethanol during adolescence (HFDc-S and HFDb-S) also exhibited higher levels of striatal CX3CL1. Again, CX3CL1 is confirmed as a more sensitive biomarker to ethanol and HFD-induced neuroinflammation. As described before, CX3CL1 has been implicated on the development of obesity-induced inflammation [124], and our results further demonstrate that bingeing or continuous access to HFD is capable of increasing neuroinflammatory signaling.

Taken together, these data indicate that early exposure to BD produces a persistent neuroinflammatory response in brain areas of the reward circuitry, which remains long after the intake of alcohol and extends into adulthood, in addition to producing an increase in the consumption of alcohol. The effect of a HFD on the increase in alcohol consumption induced by adolescence BD seems to depend on the administration pattern, with increases in alcohol intake when bingeing on HFD but decreases when HFD is continuously available. However, both types of HFD intake potentiate the alcohol-induced inflammatory profile. Consumption of intermittent or continuous HFD and alcohol has combinatorial effects on the overall inflammatory profile, with these results pointing to the immune system as a potential target to decrease vulnerability to alcohol abuse. It is important to remark that this study was performed on male mice. In order to cover the reported sex differences in the metabolism and action of alcohol and HFD, further studies should be performed on female mice. Alcohol intake damages the intestinal barrier and could produce alterations to the gut microbiota composition, which in combination with HFD could impair the intestinal tissue, promoting peripheral and central inflammation, as we observed in the present study. Although further studies are needed to continue discovering the role of diet as a vulnerability factor to the long-lasting effects caused by BD and the role of the gut–brain axis when alcohol use and a HFD are combined, the present results reveal the dangerous combination of an early exposure to ethanol and HFD intake.

## 4. Materials and Methods

### 4.1. Animals and Experimental Procedure

Adolescent male OF1 mice (*n* = 240) were ordered from Charles River (Barcelona, Spain). Animals arrived on postnatal day (PND) 21 and were housed in groups of 4 under standard conditions (cage size: 28 × 28 × 14.5 cm) with a reverse 12 h light/dark cycle (lights on at 07:30 p.m.) and constant temperature 21 ± 1 °C. All procedures were conducted in compliance with the guidelines of the European Council Directive 2010/63/EU 2017/VSC/PEA/00204 2017/VSC/PEA/00204 (11 Novemver 2017) regulating animal research and were approved by the local ethics committees (University of Valencia). Three separate cohorts of animals were used for studying the neuroinflammatory effects at different time points: immediately after the BD procedure (*n* = 12 mice/group), three weeks after the BD procedure (*n* = 12 mice/group) and after the ethanol self-administration (*n* = 10–16 mice/group).

### 4.2. Feeding Conditions

Regular chow (Teklad Global Diet 2014, 13 Kcal % fat, 67 Kcal % carbohydrates and 20% Kcal protein; 2.9 kcal/g; no sugars added) and HFD chow (TD.06415, 45 Kcal% fat, 36 Kcal % carbohydrates and 19% Kcal protein; 20% of carbohydrates are sucrose; 4.6 kcal/g) were used in our study. All mice were fed only the regular chow for a 5 day acclimation period prior to initiating the experimental feeding schedule.

In Experiment 1, at PND 26, mice were randomly assigned to continue on a (1) SD group with ad libitum access to regular chow or switch to a (2) high-fat diet binge (HFDb) group with ad libitum access to regular chow and time-restricted access to the HFD chow 3 days per week (Monday, Wednesday and Friday) to induce the development of binge-like eating [131]. For the binge session, mice were separated in individual cages and given access to pre-weighed HFD pellets for two hours. The food intake in a binge session was calculated by subtracting the weight of the remaining food pellets from the initial weight. Binge sessions took place 2 h after initiation of the dark phase.

In Experiment 2, at PND 26, mice were randomly assigned to either a (1) SD group with ad libitum access to regular chow or a (2) a continuous HFD group (HFDc) with ad libitum access to HFD chow. The animal’s body weight and home-cage intake were measured twice per week throughout the procedure.

In Experiment 1, when the SA procedure started, mice only had access to food for 1 h. In Experiment 2 mice had continuous access to water and food, except during the SA session (Figure 7).

### 4.3. Drugs

For the BD procedure, ethanol was administered intraperitoneally (i.p.) at 1.25 g/kg and diluted in 0.9% NaCl at a volume of 0.01 mL/g. The control group was injected with a physiological saline vehicle (NaCl 0.9%) used to dissolve the drugs. For the oral self-administration, absolute ethanol (Merck, Madrid, Spain) was dissolved in water using a *w*/*v* percentage, i.e., a 6% (*w*/*v*) ethanol solution equivalent to a 7.6% (*v*/*v*) ethanol solution. Saccharin sodium salt (Sigma, Madrid, Spain) was diluted in water.

### 4.4. Binge Drinking Protocol in Adolescent Mice

Animals received 16 doses of EtOH (Merck, Madrid, Spain) (1.25 g/kg) or saline over a 2 week period according to the following schedule: twice daily administrations (with a 4-h interval) on two consecutive days separated by a two-day gap in which no injections were administered [67]. Animals were injected on PND 29, 30, 33, 34, 36, 37, 38, and 39.

### 4.5. Oral Ethanol Self-Administration

This procedure is based on the one employed by Navarrete et al. [132]. Mice were tested in 8 modular operant chambers (MED Associated Inc., Georgia, VT, USA) placed inside noise isolation boxes. Each chamber was equipped with a house light, two nose-poke holes, one receptacle to deliver a liquid solution, one syringe pump, and one stimulus light and buzzer. A response on the active nose-poke resulted in the delivery of 36μL of fluid paired with a 0.5 s stimulus flash of light and a 0.5-s buzzer beep, which was followed by a 6-s time-out period. Packing software (Cibertec, SA, Spain) controlled the stimulus events and fluid delivery and recorded operant responses.

#### 4.5.1. Training Phase (8 Days)

Mice were trained to respond to the active nose-poke to receive 36 μL of 0.2% (*w*/*v*) saccharin reinforcement in 60 min sessions. In Experiment 1, to prompt the acquisition of behavior, access to the home cage chow was restricted to 1h per day two days before the initiation of the procedure. Before the first training session, water was withheld for 24 h, and access to food was provided one hour prior to the session. During the subsequent 3 days, water was provided ad libitum, except during the 1h period of food access before beginning each session, in which the water bottle was removed from the cages (postprandial). For the following four days and until the end of the procedure, water was available ad libitum and food access was provided for 1 h after the session. The feeding schedule maintained the weight at 80% of the pre-training body weight. In Experiment 2, food and water were available ad libitum throughout the procedure.

#### 4.5.2. Saccharin Fading (9 Days)

The saccharin concentration was gradually decreased as the EtOH concentration was gradually increased [133,134]. Each solution combination was set up to three consecutive sessions per combination (0.15% Sac −2% EtOH; 0.10% Sac −4% EtOH; 0.05% Sac −6% EtOH).

#### 4.5.3. Ethanol Consumption 6% (11 Days)

The aim of the last phase was to evaluate the number of responses on the active nose-poke and the 6% EtOH (*w*/*v*) consumption. During 5 daily consecutive test sessions the number of active responses and EtOH consumption (μL) was measured under a fixed-ratio (FR)-1 followed by another five consecutive days under a FR3-schedule (three responses on the active nose-poke were needed to obtain one reinforcement). Alcohol intake was determined by subtracting the alcohol leftover in the receptacle collected with a micropipette after each session.

#### 4.5.4. Progressive Responding Ratio for Alcohol

In the last session we tested the motivation to seek alcohol by setting a progressive ratio (PR) paradigm. In the PR-schedule the response requirement to obtain a reinforcement escalated according to the following series: 1-2-3-5-12-18-27-40-60-90-135-200-300-450-675-1000. The breaking point for each animal was defined as the highest number of nose-pokes each animal performed to earn one reinforcement during the 2 h session.

### 4.6. Tissue Sampling

To obtain tissue samples, mice were sacrificed by cervical dislocation 24 h after the ethanol consumption and 48 h after the binge session. Brains were rapidly removed, and the striatum was dissected out in both hemispheres following the procedure described by Heffner et al. [135], after which it was flash frozen in dry ice until storage at −80 °C.

### 4.7. IL-6 and CX3CL1 Measurements

Frozen brain striatal nuclei were homogenized in 250 mg of tissue/0.5 mL of cold lysis buffer (1% NP-40, 20 mM Tris-HCl pH 8, 130 mM NaCl, 10 mM NaF, 10 μg/mL aprotinin, 10 μg/mL leupeptin, 40 mM DTT, 1 mM Na3VO4, and 10 mM PMSF). Brain homogenates were kept on ice for 30 mim. and centrifuged at maximum speed for 15 min, after being determined by the Bradford assay from ThermoFisher (Waltham, MA, USA). Striatal IL-6 and CX3CL1 concentrations were quantified by using an enzyme-linked immunosorbent essay (Mouse IL-6 ELISA Kit, ab 100712; Mouse Fractalkine ELISA Kit, ab100683) following the manufacturer’s protocol (Abcam, Cambridge, UK). To determine absorbance, we employed an iMark microplate reader (Bio-RAD, Hercules, CA, USA) controlled by Microplate Manager 6.2 software. The optical density was read at 450 nM and the final results were calculated using a standard curve carried according to the manufacturer’s instructions. The data were expressed as pg/mL for plasma, and pg/mg for tissue samples.

### 4.8. Statistical Analyses

In both experiments, data relating to body weight was analyzed by a two-way ANOVA with a two between-subject’s variable -Treatment (saline and ethanol) and Diet (standard diet (SD) and continuous high-fat diet (HFDc) or binge high-fat diet (HFDb))- and a within variable Weeks with 10 levels (from PND 25 to 88). The total weekly energy intake was analyzed by a two-way ANOVA with two between variables -Treatment (saline and ethanol) and Diet (standard diet (SD) and continuous high-fat diet (HFDc) or binge high-fat diet (HFDb)). Intake during bingeing sessions was analyzed by a one-way ANOVA with one between-subject’s variable—Treatment (saline and ethanol) and a within variable Weeks with 10 levels.

In Experiment 1, EtOH SA and effective responses were analyzed individually by a two-way ANOVA with a two between-subject’s variable -Treatment (saline and ethanol) and Diet (standard diet (SD), high-fat binge (HFDb)- and a within-subject’s variable—Days, with five levels of FR1 or FR3 schedule. In addition, differences between the saline groups in the FR1 were tested with a two-way ANOVA with one between-subject’s variable Diet (standard diet (SD), high-fat binge (HFDb)- and a within-subject’s variable—Days, with five levels of FR1. A two-way ANOVA with a two between-subject’s variable—Treatment (saline and ethanol) and Diet (standard diet (SD), high-fat binge (HFDb)) was employed to analyze ethanol consumption and breaking point values during the PR session.

In Experiment 2, two two-way ANOVAs were performed with a two between-subject’s variable—Treatment (saline and ethanol) and Diet (standard diet (SD), continuous high-fat diet (HFDc)- and a within-subject’s variable -Days, with five levels of FR1 or FR3 schedule. A two-way ANOVA with a two between-subject’s variable—Treatment (saline and ethanol) and Diet (standard diet (SD), continuous high-fat diet (HFDc)) was employed to analyze ethanol consumption and breaking point values during the PR session.

A two-way ANOVA was employed to analyze the data from the biochemical parameters (ELISA assays for the CX3CL1 and IL-6) with a two between-subject’s variable, previously described, Treatment and Diet. For the data obtained after the self-administration procedure, two different ANOVAs were performed for Experiment 1 and 2. In all cases, post-hoc comparisons were performed with Bonferroni tests. All statistical analyses were performed using SPSS Statistics v26. Data were expressed as mean ± SEM and a value of *p* < 0.05 was considered statistically significant.

## Figures and Tables

**Figure 1 ijms-22-05279-f001:**
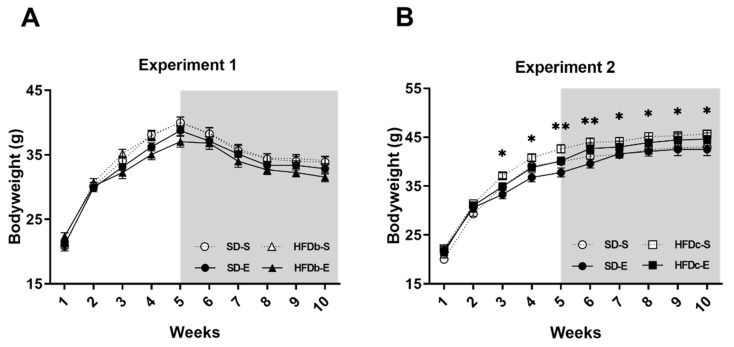
Body weight of mice over the procedure. Mean (± SEM) amount measured animals’ body weight. (**A**) No differences in body weight between the standard diet (SD) and high-fat diet binge (HFDb) groups were detected; (**B**) Significant differences between continuous high-fat diet (HFDc) groups and SD groups. * *p* < 0.05; ** *p* < 0.01.

**Figure 2 ijms-22-05279-f002:**
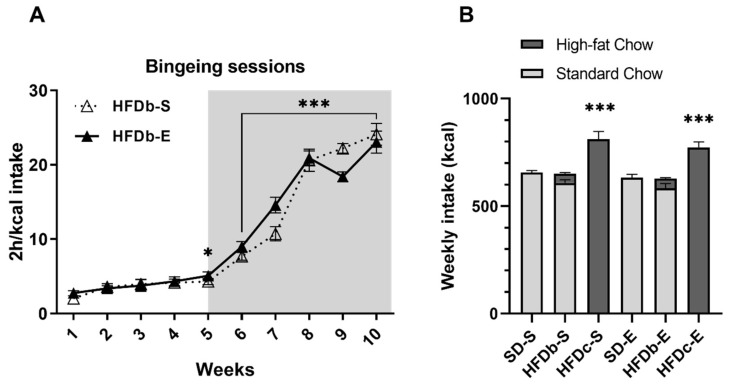
(**A**) Binge sessions. Caloric intake (kcal) of high-fat diet (HFD) in the 2-h high-fat binge-eating sessions that took place on Monday, Wednesday and Friday in the high-fat diet binge (HFDb) groups. The mean (±SEM) amount of kcal consumed in 2 h of access to high-fat food in every binge session * *p* < 0.05; *** *p* < 0.001 significant difference with respect to the first and second binge sessions. (**B**) Weekly energy consumption per cage (four mice) across the procedure. Mean (±SEM) 24 h energy consumption (kcal) of mice (standard chow shown by open bars and HFD by solid bars) *** *p* < 0.001 significant difference of both continuous high-fat diet (HFDc) groups with respect to the standard diet (SD) groups and HFDb groups.

**Figure 3 ijms-22-05279-f003:**
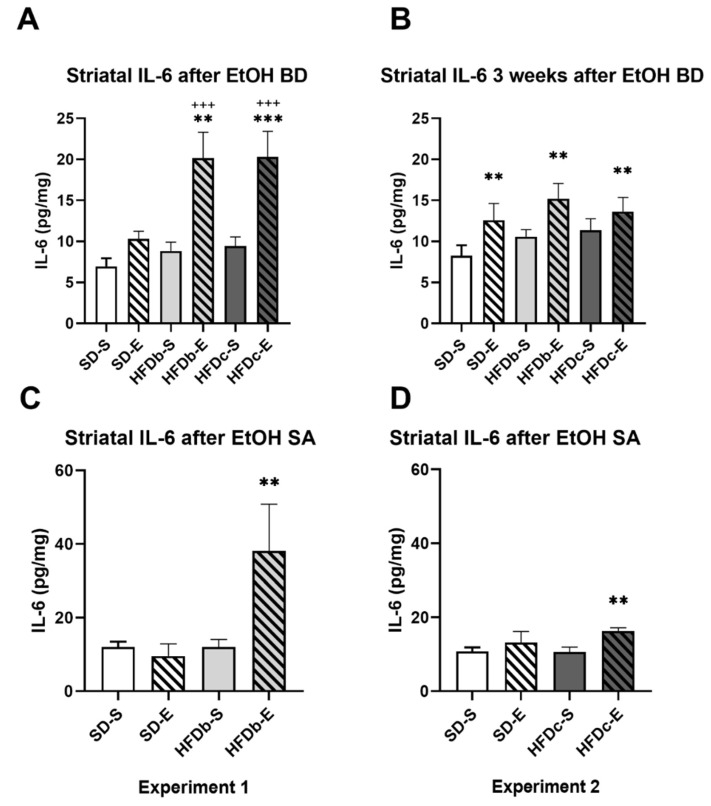
Intermittent ethanol intake during adolescence and high-fat diet (HFD) increase IL-6 levels in the striatum. The columns represent the mean and the vertical lines ± SEM of concentration levels of IL-6 (pg/mg protein) (**A**) after ethanol binge drinking (BD) during adolescence, (**B**) 3 weeks after the last ethanol injection and (**C**) after the end of the oral self-administration (SA) procedure in the groups with food deprivation and (**D**) in the groups without food deprivation *** *p* < 0.001, ** *p* < 0.01 with respect to the corresponding non-ethanol-treated groups; +++ *p* < 0.001 with respect to the corresponding standard diet fed group.

**Figure 4 ijms-22-05279-f004:**
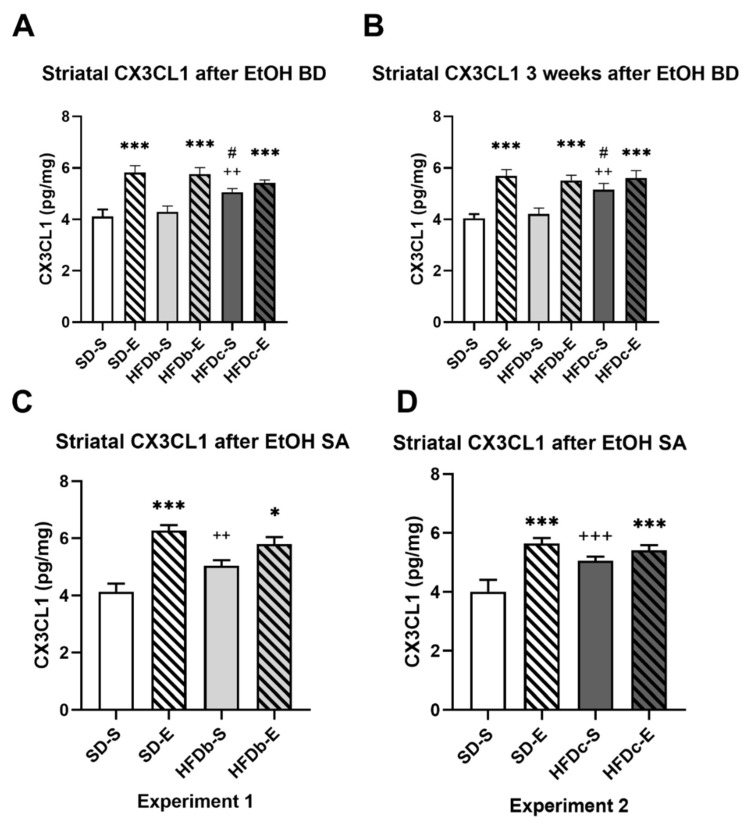
Intermittent ethanol intake during adolescence and continuous high-fat diet (HFD) increase CX3CL1 levels in the striatum. The columns represent the mean and the vertical lines ± SEM of concentration levels of CX3CL1 (ng/mg protein) (**A**) after ethanol BD during adolescence, (**B**) 3 weeks after the last ethanol injection and (**C**) after the end of the oral self-administration (SA) procedure in the groups with food deprivation and (**D**) in the groups without food deprivation *** *p* < 0.001, * *p* < 0.05 with respect to the corresponding non-ethanol-treated groups; +++ *p* < 0.001, ++ *p* < 0.01 with respect to the corresponding standard diet fed group. # *p* < 0.05 with respect to the corresponding binge HFD group.

**Figure 5 ijms-22-05279-f005:**
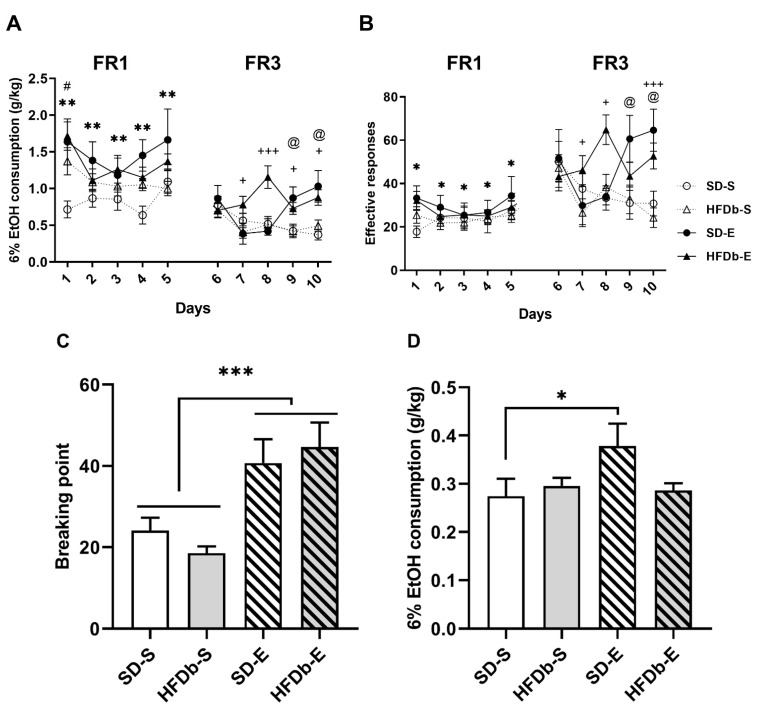
Effects of intermittent high-fat diet (HFD) bingeing on effective responses and oral ethanol EtOH self-administration in OF1 mice. Data are presented as mean (± SEM): (**A**) g/kg of 6% EtOH consumption and (**B**) the number of effective responses during FR1 and FR3. The columns represent means and the vertical lines ± SEM of (**C**) breaking point values during PR and (**D**) amount of 6% EtOH consumption during the PR session. * *p* < 0.05; ** *p* < 0.01; *** *p* < 0.001 values that are significantly different from animals that received BD compared to the saline groups. + *p* < 0.05; +++ *p* < 0.001 significant differences in the HFDb-E compared to the HFD-S group. @ *p* < 0.05 significant differences in SD-E group with respect to SD-S. # *p* < 0.05 significant differences in the HFDb group between Day 1 compared to Day 2, 3 and 4.

**Figure 6 ijms-22-05279-f006:**
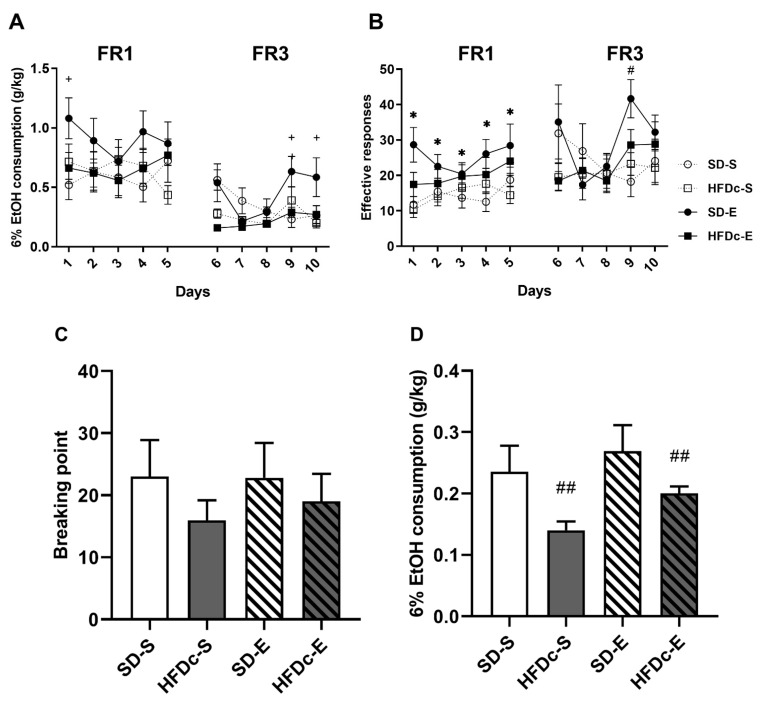
Effects of continuous access to high-fat diet (HFD) on effective responses and oral ethanol EtOH self-administration in OF1 mice. Data are presented as mean (± SEM): (**A**) g/kg of 6% EtOH consumption (**B**) the number of effective responses during FR1 and FR3. The columns represent means and the vertical lines ± SEM of (**C**) breaking point values during PR and (**D**) amount of 6% EtOH consumption during the PR session * *p* < 0.05 significant differences in the groups that received BD (SD-E and HFD-E) compared to the saline groups (SD-S and HFDc-S). + *p* < 0.05 significant differences between SD-E with respect to SD-S. # *p* < 0.05 significant differences between the BD-treated groups and the saline-treated groups in Day 9. ## *p* < 0.01 significant differences between the HFDc groups (HFDc-S and HFDc-E) compared to the SD groups (SD-S and SD-E).

**Figure 7 ijms-22-05279-f007:**
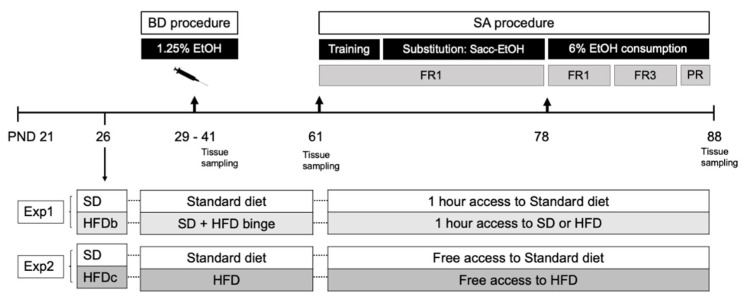
Schematic representation of experimental design. OF1 adolescent mice arrived in the laboratory at PND 21. At PND 26 all animals changed their feeding schedules according to the experimental condition (1) Standard diet (SD), (2) High-fat diet binge (HFDb) or (3) High-fat diet continuous access (HFDc). From PND 29 to 41 mice underwent the (1.25% ethanol) binge drinking procedure (BD) (SD-E, HFDb-E and HFDc-E) or were treated with saline (SD-S, HFDb-S, HFDc-S). After reaching adulthood (PND 61), mice started the self-administration (SA) procedure. Tissue sampling was performed at three different timepoints: after BD, three weeks after BD and after the SA procedure.

## Data Availability

Not applicable.

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
