# Peer review of "Pairing Binge Drinking and a High-Fat Diet in Adolescence Modulates the Inflammatory Effects of Subsequent Alcohol Consumption in Mice"

_ijms, 2021, doi:10.3390/ijms22105279_

Round 1

Reviewer 1 Report

The manuscript presents a study of pairing binge drinking and a high-fat diet during adolescence and its association to inflammatory markers and alcohol drinking in adulthood on an animal model. The authors performed well-designed experiments and showed that mouse bodyweight is increased by continuous but not intermittent access to a high-fat diet, intermittent access to a high-fat diet induces bingeing behavior.  The measurement of the IL-6 and CX3CL1 cytokine levels at three different time points showed that a high-fat diet and intermittent ethanol intake during adolescence increases neuroinflammatory response.  The authors concluded that adolescent binge drinking and intermittent high-fat diet increase adult alcohol intake and pointed to neuroinflammation as an important mechanism modulating this interaction. The results are clearly presented and the conclusions are well supported.

Author Response

We appreciate Reviewer’s comments on our manuscript.

Reviewer 2 Report

The methods are sound and the statistical methods used seem appropriate.  There are only a few concerns as listed below.

In the abstract and in particular in the introduction a disproportionate amount of time is spent on the gut microbiome, but there were no gut microbiota manipulations and no markers of gut microbiome were measured. In the discussion it is also mentioned. It is very curious why it is being “sold” as a gut microbiota paper.

Many of the phenomena have already been demonstrated in previous work by this group and others. The work is primarily descriptive and adds incrementally to the body of knowledge.

Why only measure inflammatory markers in the striatum? There may be differences in the inflammatory response in other regions of the brain, and no real rationale for the exclusive interest in the striatum was provided.  Furthermore, the sentence “we measured two inflammatory markers in the striatum when these two factors were combined.” does not make sense.

Author Response

1- In the abstract and in particular in the introduction a disproportionate amount of time is spent on the gut microbiome, but there were no gut microbiota manipulations and no markers of gut microbiome were measured. In the discussion it is also mentioned. It is very curious why it is being “sold” as a gut microbiota paper.

We offer a brief description of the relationship between the microbiota and the consumption of high-fat diet and alcohol because it is one of the major mechanisms triggering inflammation. We appreciate your suggestion and accordingly, we have reduced the volume of the text devoted to the gut microbiome in the discussion. Finally, it is important to remark that this article was submitted in response to an invitation to the special issue of International Journal of Molecular Sciences “Gut Microbiota and Immunity”.

2-Many of the phenomena have already been demonstrated in previous work by this group and others. The work is primarily descriptive and adds incrementally to the body of knowledge.

Certainly, our research group has been studying multiple dimensions of alcohol consumption during adolescence. In this work, we offer for the first time an account of the inflammatory effects of the combined consumption of alcohol and high-fat diet. As we described in the text, we included two different high-fat diet feeding schedules to test if the pattern of consumption had an impact on the inflammatory markers. To our knowledge there is no available evidence comparing these two patterns of HFD intake  with alcohol exposure during adolescence. In addition, no study has measured IL-6 nor CX3CL1 levels after a oral alcohol self-administration procedure.

Our study offers a full-fledged model of the nutritional habits and pattern of alcohol consumption among adolescents. One particular strength of this study lies on the combination of behavioral and biochemical measures. The alcohol self-administration offers information about the behavioral consequences of binge drinking and intermittent or continuous HFD consumption during adolescence. These results are combined with biochemical analyses that measure the resulting inflammatory profile at adolescence, during adulthood and after the self-administration.

3- Why only measure inflammatory markers in the striatum? There may be differences in the inflammatory response in other regions of the brain, and no real rationale for the exclusive interest in the striatum was provided. 

The striatum is a key structure in the reward circuitry modulating dopaminergic signaling. In the case of alcohol, alcohol dependence has been associated with a blunted dopamine transmission in the ventral striatum. Systematically, research has reported increased inflammatory activity in the striatum after alcohol exposure (Pascual et al., 2015). More specifically, IL-6 signaling in the microglia reduces the excitability of striatal neurons (Klawonn et al., 2021). Furthermore, evidence has shown that the gut microbiome of compulsive alcohol drinkers is correlated with abnormal striatal dopamine signaling (Jadhav et al., 2018).

4- Furthermore, the sentence “we measured two inflammatory markers in the striatum when these two factors were combined.” does not make sense.

We agree with the reviewer that this sentence is not clear. We aim to mean that both HFD and ethanol induces neuroinflammatory processes in overlapping brain circuits, therefore we measured two inflammatory markers in the striatum when their consumption is combined during adolescence.

To clarify this point, the sentence has been rewritten as follow:

Considering that both HFD and ethanol induces neuroinflammatory processes in overlapping brain circuits, we measured two inflammatory markers in the striatum when their consumption is combined.

Reviewer 3 Report

The manuscript under review provides information on the neuroinflammatory response of adolescent binge drinking when combined with a continuous or intermittent high-fat diet and its effects on adult ethanol consumption. Overall, the manuscript is well-written and organized, although it is not always easy to follow due to the wide variety of treatments and the use of abbreviations. The results are very interesting, but it is a pity that the authors have not analysed the gut microbiota of the mice in study to better understand the role of the HFA and alcohol in the microbiota and their effect in the neuroinflammatory response. I think it is also important for the authors to mention throughout the manuscript that the results present in this manuscript are based on male mice. This manuscript may be sexually biased and the results may be different in females, as many neurological and physiological differences have been found in different sexes. For example, the dopamine system is different in males and females.

Finally, my main concern is the statistical analysis carried out in the study. It seems to me that animals in the same cage are treated as independent of each other, and the authors do not control the cage effect. This is really important because even if you do not look at the gut microbiota in your study, you mention it as a possible factor influencing neuroinflammation. We know that the cage effect can have a huge effect on the composition of the gut microbiota, so it must be controlled.

Below I provide other comments and suggestions that the authors might consider useful for a possible new version of the manuscript.

Abstract:

Line 21: I would add that the experiment was performed with mice. E.g. … using a self-administration (SA) paradigm in mice.

Results

L132: Since you only mentioned what SA means in the abstract, I would change “SA deprivation” to “self-administration (SA) deprivation”.

L135: p<0.05 instead of p<0,05

L138: Remove the comma after (SD-S and SD-E)

L142: Full stop, or colon or semicolon (and small letter) after “weight”

L158: 2h instead of 2 h, write it the same way throughout the text

Discussion

L134: Make it clear that you are working with mice because adolescence and adulthood are used in humans as well and it can be confusing

L342: I think this is the first time you mention BED patient. What does it mean? Make sure the text in understandable without having to check it on the original paper.

L385: It is not necessary to place a comma after Mazzone et al.

L433: Remove brackets after [62,111]

L433: After “Long-term HFD feeding causes a microbiota modification, changing intestinal permeability and increasing bacterial LPS” a reference is necessary, since the gut microbiota was not analyse in this study. Also, write what LPS stands for.

L437: Eliminate the coma after [21

Material and Methods

L517: Did the mice have enrichment material in the cage?

L524: I would recommend you to write the entire word the first time you mention it in the M&M.

L516: Include how many animals were purchased in total.

526: I assume that n=12 mice/group means 3 cages (with 4 animal in each cage), but it is not clear to me what 10-16 mice/group means.

Figure 7. Explain further the schematic representation of the experimental design. The reader must be able to understand it without the need of reading the entire manuscript. You have included a lot of abbreviation without explanation. Also change 1,25% to 1.25%.

L531: Write 2.9

L531: Is TD.06415 also Teklad diet?

L532: In HFD chow you do not specify how many kcal/g

L533: It is a repeat. Decide if you want to include here or in the previous paragraph and delete the other

L535: Figure 7 makes me understand that is PND 24 instead of 26. Make it clear.

L535: Were the mice assigned randomly to different treatment, or were the cages?

L542:  What were the conditions for working in dark phase? Did you turn on the lights, use red lights…?

L544: It is not clear to me why you wrote (1) and (3), and there is no (2)

L552: Is it the SD group? Write the group name to avoid confusions

L551: Add a reference for the ethanol, as you did with the absolute ethanol

L556: What did you use for controls in the oral self-administration experiment?

L556: Briefly explain what the saccharin sodium salt was used for

Author Response

1- The results are very interesting, but it is a pity that the authors have not analysed the gut microbiota of the mice in study to better understand the role of the HFA and alcohol in the microbiota and their effect in the neuroinflammatory response.

Certainly, we agree that measuring some parameters of the gut microbiome could have enriched our work. Unfortunately, it has not been the case for this work. For future studies we will strongly consider including this important source information.

2- I think it is also important for the authors to mention throughout the manuscript that the results present in this manuscript are based on male mice. This manuscript may be sexually biased and the results may be different in females, as many neurological and physiological differences have been found in different sexes. For example, the dopamine system is different in males and females.

Definitely, we have remarked in the text the importance of testing female mice in order to find sex differences.

It is important to remark that this study was performed on male mice. In order to cover the reported sex differences in the metabolism and action of alcohol and HFD, further studies should be performed on female mice. (Line 528)

3- Finally, my main concern is the statistical analysis carried out in the study. It seems to me that animals in the same cage are treated as independent of each other, and the authors do not control the cage effect. This is really important because even if you do not look at the gut microbiota in your study, you mention it as a possible factor influencing neuroinflammation. We know that the cage effect can have a huge effect on the composition of the gut microbiota, so it must be controlled.

As the Reviewer’s remarks, the microenvironment (cage effect) is an important factor that affects the gut microbiota population (Deloris et al., 2006), and thus has important consequences for experimental design. We have to point out that our experiment was not designed for study microbiome composition. Therefore, we did not take into consideration the cage effect. We hypothesized that the increased neuroinflammatory response could be mediated by changes in gut microbiome. In our study, every animal in the same cage belonged to the same experimental condition. The aim of our study was to determine the effects of two different high-fat diets on neuroinflammatory signaling and alcohol intake in mice exposed to binge drinking during adolescence. We predicted that different diets would influence the gut microbiota differentially and thus, would result in a different inflammatory profile. We assume that co-housing 4 mice in each cage would resulted in a synchronized microbiota but the same diet could induce the same microbiome effects. However, if determination of gut microbiome were performed in the study, we had equalized the microbiota across experimental conditions with mixed environment or isolation of the animals.

He leído que lo que recomiendan para difuminar el efecto de la sincronización de la microbiota es el mixed environment (los grupos experimentales mezclados en las cajas o que estén alojados individualmente).

4- Below I provide other comments and suggestions that the authors might consider useful for a possible new version of the manuscript.

Abstract:

Line 21: I would add that the experiment was performed with mice. E.g. … using a self-administration (SA) paradigm in mice.

Results

L132: Since you only mentioned what SA means in the abstract, I would change “SA deprivation” to “self-administration (SA) deprivation”.

L135: p<0.05 instead of p<0,05

L138: Remove the comma after (SD-S and SD-E)

L142: Full stop, or colon or semicolon (and small letter) after “weight”

L158: 2h instead of 2 h, write it the same way throughout the text

We appreciate Reviewer’s comments and suggestions to improve our manuscript. All of these suggestions have been corrected in the text.

Discussion

L134: Make it clear that you are working with mice because adolescence and adulthood are used in humans as well and it can be confusing

We agree with the Reviewer’s remark and thus, it is has been added to the text as follows:

We aimed to test the effects of HFD consumption on two important associated consequences of early BD: the neuroinflammatory response and the increased alcohol drinking in adult mice. (Line 332)

L342: I think this is the first time you mention BED patient. What does it mean? Make sure the text in understandable without having to check it on the original paper.

BED stands for binge eating disorder. The topic and abbreviation are included in the introduction (Line 80).

L385: It is not necessary to place a comma after Mazzone et al.

This typing error has been corrected. (Line 406)

L433: Remove brackets after [62,111]

L433: After “Long-term HFD feeding causes a microbiota modification, changing intestinal permeability and increasing bacterial LPS” a reference is necessary, since the gut microbiota was not analyse in this study. Also, write what LPS stands for.

We wish to thank you for the suggestion. We have included in the text: lipopolysaccharide (LPS) [112].

The reference that has been added is the following:

Zhang, M., & Yang, X. J. (2016). Effects of a high fat diet on intestinal microbiota and gastrointestinal diseases. World journal of gastroenterology22(40), 8905.

L437: Eliminate the coma after [21

Material and Methods

L517: Did the mice have enrichment material in the cage?

Male mice in regular housing condition were housed in groups of four in transparent plastic cages (27 × 27× 14 cm) with no more enrichment than standard bedding (wood flakes 1-3.35 mm), nesting material (paper strands) and two wooden gnaw sticks (5 x 1 x 1 cm) per cage.

L524: I would recommend you to write the entire word the first time you mention it in the M&M.

L516: Include how many animals were purchased in total.

As the Reviewer suggested we have included in the text the total number of animals used in the study.

Adolescent male OF1 mice (N=240) were ordered from Charles River (Barcelona, Spain). Line 539

For this study, the total number of animals used was 240.

526: I assume that n=12 mice/group means 3 cages (with 4 animals in each cage), but it is not clear to me what 10-16 mice/group means.

As the Reviewer assumes, n=12 mice/group consists of 3 cages with 4 animals in each cage. In the case of 16 mice/group consists of 4 cages with 4 animals in each cage. It is important to remark that after the training phase in the self-administration procedure some animals were not included in the study because they did not fulfill the learning acquisition criteria.

Figure 7. Explain further the schematic representation of the experimental design. The reader must be able to understand it without the need of reading the entire manuscript. You have included a lot of abbreviation without explanation. Also change 1,25% to 1.25%.

For a better understanding of our experimental design we have included an explanatory text for the Figure 7.

Figure 7. Schematic representation of experimental design. OF1 adolescent mice arrived at postnatal day (PND) 21 to the laboratory. At PND 26 all animals change their feeding schedules according to the experimental condition (1) Standard diet (SD), (2) High-fat diet binge (HFDb) or (3) High-fat diet continuous access (HFDc). From PND 29 to 41 mice underwent the (1.25% ethanol) binge drinking procedure (BD) (SD-E, HFDb-E and HFDc-E) or were treated with saline (SD-S, HFDb-S, HFDc-S). After reaching adulthood (PND 61) mice started the self-administration (SA) procedure. Tissue sampling was performed at three different timepoints; after BD, three weeks after BD and after the SA procedure. 

L531: Write 2.9

L531: Is TD.06415 also Teklad diet?

TD refers to the reference of the high-fat diet.  

L532: In HFD chow you do not specify how many kcal/g

This piece of information has been added to the manuscript.

Regular chow (Teklad Global Diet 2014, 13 Kcal % fat, 67 Kcal % carbohydrates and 20% Kcal protein; 2.9 kcal/g; no sugars added) and HFD chow (TD.06415, 45 Kcal% fat, 36 Kcal % carbohydrates and 19% Kcal protein; 20% of carbohydrates are sucrose; 4.6kcal/g) were used in our study.

L533: It is a repeat. Decide if you want to include here or in the previous paragraph and delete the other

As the Reviewer noticed, this information was redundant. We have maintained the sentence on line 563.

L535: Figure 7 makes me understand that is PND 24 instead of 26. Make it clear.

In accordance with the Reviewer´s suggestion, we apologize for the mistake and modify the figure. It has been changed in the timeline. The feeding condition is established at PND 26.

L535: Were the mice assigned randomly to different treatment, or were the cages?

The cages (4 mice each) were randomly assigned to different treatments. In this way, all animals in a cage belonged to the same experimental condition.

L542:  What were the conditions for working in dark phase? Did you turn on the lights, use red lights…?

Yes, due to the reverse of the light cycle, in our laboratory we use red light. As described in the text: with a reverse 12-h light/dark cycle (lights on at 07:30 p.m.)

L544: It is not clear to me why you wrote (1) and (3), and there is no (2)

In the study there are only three feeding conditions (1) SD, (2) HFDb and (3) HFDc. The (1) condition is identical in both experiments. We agree with the Reviewer in considering the listing confusing. We have changed the listing numbers so that Experiment 2 has two experimental conditions (1) and (2).

In Experiment 1, at PND 26, mice were randomly assigned to continue on a (1) SD group with ad libitum access to regular chow or switch to a (2) high-fat diet binge (HFDb) group with ad libitum access to regular chow and time-restricted access to the HFD chow 3 days per week (Monday, Wednesday and Friday) to induce the development of binge-like eating [132]. For the binge session, mice were separated in individual cages and given access to pre-weighed HFD pellets for two hours. The food intake in a binge session was calculated by subtracting the weight of the remaining food pellets from the initial weight. Binge sessions took place 2 hours after initiation of the dark phase.

In Experiment 2, at PND 26, mice were randomly assigned to either a (1) SD group with ad libitum access to regular chow or a (2) a continuous HFD group (HFDc) with ad libitum access to HFD chow. The animal’s body weight and home-cage intake were measured twice per week throughout the procedure.

L552: Is it the SD group? Write the group name to avoid confusions

It has been added to the manuscript.

L551: Add a reference for the ethanol, as you did with the absolute ethanol. The same ethanol was used in both protocols.

We used the same alcohol for both protocols, the self-administration procedure and the binge drinking protocol. We add again the reference to the text for the latter. (Line 590)

L556: What did you use for controls in the oral self-administration experiment?

The control group in the self-administration experiment are the SD-S group. Mice in this group received saline treatment during adolescence and were fed ad libitum with standard diet.

L556: Briefly explain what the saccharin sodium salt was used for

Saccharin sodium salt is used for the solution used during the training phase of the self-administration procedure. In order to overcome the alcohol taste aversion in mice, saccharin fading adds sweet flavour to the solution and thus becomes less aversive.

As it is described in the text:  The saccharin concentration was gradually decreased as the EtOH concentration was gradually increased [133,134]. Each solution combination was set up to three consecutive sessions per combination (0.15% Sac -2% EtOH; 0.10% Sac -4% EtOH; 0.05% Sac -6% EtOH).

Round 2

Reviewer 3 Report

The authors have added most of my comments and suggestions to the manuscript correctly. Just one small thing that I think is important to ensure the reproducibility of the study is to include the information about the enrichment used in the cages that the authors provided in the author's response document.

Best wishes